# Antiproliferative and Enzyme Docking Analysis of Engleromycin from *Engleromyces goetzei*

**DOI:** 10.3390/molecules24010166

**Published:** 2019-01-04

**Authors:** Yongli Zhang, Guilin Chen, Hong Ma, Mingquan Guo

**Affiliations:** 1Key Laboratory of Plant Germplasm Enhancement and Specialty Agriculture, Wuhan Botanical Garden, Chinese Academy of Sciences, Wuhan 430074, China; lisy19881120@163.com (Y.Z.); glchen@wbgcas.cn (G.C.); hongma01@yahoo.com (H.M.); 2Sino-Africa Joint Research Center, Chinese Academy of Sciences, Wuhan 430074, China; 3Haematology and Oncology Division, Children’s Hospital Los Angeles, Los Angeles, CA 90027, USA

**Keywords:** *Engleromyces goetzei*, engleromycin, antiproliferative activities, molecular docking

## Abstract

*Engleromyces goetzei* P. Henn. (*E. goetzei*) has been widely used as a traditional herb for many years in Kenya due to its diverse biological effects. Although engleromycin was first isolated from *E. goetzei* in 1980, its pharmacological activity is still unknown. In this study, engleromycin from *E. goetzei* was identified by spectroscopic analyses, and subsequently examined for its antiproliferative activity using human cancer cell lines of SGC-7901, HT-29, HeLa and A549. As a result, it was revealed that engleromycin strongly inhibited the growth of SGC-7901, HT-29, HeLa and A549 cells with IC_50_ values at 26.77 ± 1.69 µM, 7.73 ± 0.18 µM, 7.00 ± 0.12 µM and 3.14 ± 0.03 µM, respectively. The results of topoisomerase II (Top II) inhibition assay in vitro implied that engleromycin might be a Top II inhibitor. Further insights into the potential mechanism of antiproliferative activity displayed that engleromycin could dock into the binding pockets of Top II, like the clinical inhibitor doxorubicin, and then inhibit the biological activity of Top II. Taken together, our findings suggest that engleromycin has an anticancer potential, and may serve as a leading compound for the development of antitumor agents.

## 1. Introduction

Cancer is the second leading cause of death worldwide and approximately 70% of cancer deaths occurred in low and middle-income countries. Despite advances in oncology research, a thorough cure has not been found so far. Thus, there is an urgent need to develop more selective and effective anti-tumor drugs [1].

*Engleromyces goetzei* P. Henn., a species of fungus, belongs to the genus *Engleromyces*. It was described by German mycologist Paul Christoph Hennings in 1980 [2]. *Engleromyces* contains two species, and which were found in the regions of Tibet, Yunnan province of China, and Kenya highlands [3]. As a traditional African medicine, *E. goetzei* is a reputed medicine against pneumonia, colds, fever, malaria-related headache as well as liver diseases and abdominal pains [4]. *Eosimias sinensis*, collected from China, was also reported to possess a wide range of biological activities, such as anti-inflammatory, anti-microbial, anti-cancer, anti-virus, cholesterol ester transfer protein (CETP) inhibition activity [5,6,7]. Up to now, the research on *E. sinensis* in China has been recorded mostly under the name *E. goetzei.* In 1980, Pedersen first isolated engleromycin from the fruiting body of *E. goetzei* and identified it as cytochalasin [2], however, there have been no further reports on the biological activity of engleromycin.

DNA topoisomerases are essential for DNA replication, transcription, recombination, repair, and mitosis by introducing transient single-strand breaks (SSBs) or double-strand breaks (DSBs) in the DNA to adjust the topology of DNA in the cell [8,9]. Depending on the number of strands cut in one round of action, topoisomerases are separated into topoisomerase I (Top I, single strand) and topoisomerase II (Top II, double strands) [10], and they have been identified as important target enzymes for anticancer drug researches [11]. Inhibitors targeting Top II are divided into two broad classes: Top II poisons and Top II catalytic inhibitors. Top II poisons act by increasing the levels of Top II-DNA covalent complexes, leading to the accumulation of nonreversible DNA double-strand breaks. Most of the clinically active Top II inhibitors are Top II poisons, including doxorubicin, etoposide and mitoxantrone [12]. Top II catalytic inhibitors act directly on Top II or DNA, blocking the catalytic activity of Top II by preventing the Top II binding to DNA. Top II catalytic inhibitors include aclarubicin and MST-16 [13,14]. Doxorubicin is a broad-spectrum anticancer drug that targets on Top II through stabilizing the topoisomerase-DNA complex, thus leading to the accumulation of DNA breaks and eventually cancer cell death [15]. 

In this study, engleromycin was isolated from *E. goetzei* and its structure was confirmed by ^1^H-NMR and ^13^C-NMR. Then, the antiproliferative assays on human cancer cell lines of SGC-7901, HT-29, HeLa and A549 were employed to assess its cytotoxicity activity in an effort to further explain the reported anticancer activity of *E. goetzei*. Finally, Top II inhibition assay in vitro, and molecular docking simulations between engleromycin and the potential targets Top II were carried out to further explore the potential antiproliferative mechanism. 

## 2. Results and Discussion

### 2.1. Structure Identification

Compound from *E. goetzei* (EG) of EG-7h presents as white amorphous powder and its ^1^H-NMR spectrum typically gave an amide proton at δ_H_ 8.30 (1H, s, -NH), three hydroxyl protons at δ_H_ 4.38 (1H, d, *J* = 6.36, H-7), 4.32 (1H, s, H-18) and 5.29 (1H, ddd, *J* = 15.47, 10.12, 5.49, H-21), four olefinic protons at δ_H_ 4.97 (1H, s, H-12A), 4.75 (1H, s, H-12B), 5.21 (1H, d, *J* = 5.69, H-13) and 5.59 (1H, dd, *J* = 15.43, 9.63, H-14), five aromatic protons at δ_H_ 7.32-7.18 (5H, m, H-2′, 3′, 4′, 5′, 6′), three methyl groups at δ_H_ 1.36 (3H, d, *J* = 12.88, H-23), 1.03 (3H, d, *J* = 6.69, H-22) and 0.45 (3H, d, *J* = 6.68, H-11). The ^13^C-NMR of EG-7h characteristically provided one carbonyl carbon (δ_C_ 216.1, C-17), one amide carbon (δ_C_ 216.1, C-175.3), four olefinic carbon signals at δ_C_ 151.3 (C-6), 131.4 (C-14), 130.3 (C-13) and 111.0 (C-12), six aromatic carbon signals at δ_C_ 137.4 (C-1′), 130.0 (C-2′, 6′), 128.2 (C-3′, 5′) and 126.5 (C-4′), three oxygenated carbon signals at 76.6 (C-18), 72.9 (C-21) and 70.3 (C-7), and three methyl carbon signals at 23.0 (C-23), 18.9 (C-22) and 12.8 (C-11). Additionally, the complete assignments of proton and carbon signals in the structure were further confirmed by using 2D-NMR techniques, such as ^1^H-^1^H COSY, HMBC and HMQC spectra, and which indicated that EG-7h was engleromycin by comparison with the previous report [2]. In the positive ESI-MS analysis, EG-7h produced a quasi-molecular ion at *m*/*z* 482.24 ([M + H]^+^, C_28_H_36_NO_6_), and followed by the collision induced dissociation (CID) in the mass range *m*/z 50-500. Briefly speaking, the successive loss of two and three water molecules yielded the peak at *m*/*z* 446 ([M + H − 2H_2_O]^+^) and the base peak at *m*/*z* 428 ([M + H − 3H_2_O]^+^). The CID spectra at *m*/*z* 404 was observed due to the neutral loss of phenyl radical (C_6_H_6_, loss of 78 KD) from the protonated EG-7h, and which also generated a series of complicated fragments at *m/z* 386, 344, 281 and 239 through the successive loss of 18 KD (H_2_O), 42 KD (H_2_O + CO), 63 KD (H_3_O_2_ + CO) and 42 KD (C_2_H_2_ + CH_2_). Together, the ion at *m*/*z* 120 (phenylalanine radical) was obtained due to the characteristic retro-Diels–Alder (RDA) cleavage at the lactamide ring, and fragment at *m*/*z* 91 indicated the presence of the benzyl radical. Figure 1 shows the structure of engleromycin (structure **1**) and its analogues of cytochalasin D (structure **2**) from *E. goetzei* or *E. sinensis* [16,17].

### 2.2. Effects of Engleromycin on Cell Growth of SGC-7901, HT-29, HeLa, A549 and LO-2

The reported anticancer activity of *E. goetzei* prompted us to assess the cytotoxicity activity of engleromycin against human cancer cell lines. To assess the cytotoxicity of engleromycin against human cancer cells, we performed a MTT assay to detect the half inhibitory concentration (IC_50_) of engleromycin against human cancer cell lines: SGC-7901, HT-29, HeLa and A549. Meanwhile, the selectivity of engleromycin was assessed using the normal human liver cell LO-2. As shown in Table 1, engleromycin exhibited significant anti-proliferative activities against cancer cells of SGC-7901, HT-29, HeLa and A549 with IC_50_ values of 26.77 ± 1.69 µM, 7.73 ± 0.18 µM, 7.00 ± 0.12 µM, 3.14 ± 0.03 µM and 3.76 ± 0.29 µM, respectively. It is worth noting that A549 showed the highest sensitivity. We also noticed that engleromycin severely inhibited the proliferation of normal cell LO-2.

Based on the results of Figure 1, the structure of engleromycin is highly similar to cytochalasin D [2]. Cytochalasin D belongs to a large family of cytochalasins that have the ability to inhibit the polymerization of actin [18,19]. Apart from being involved in cellular morphology, cell division, cellular translocation, and apoptosis, cytochalasin D also participates in cell enucleate and protein synthesis [20]. In terms of pharmacological activity, cytochalasin D possess anti-inflammatory, antivirus, and anticancer activities [21,22,23,24,25]. Cytochalasin D can effectively induce CT26 tumor cell apoptosis and proliferation inhibition [26]. As we know, cytochalasin D acts on actin microfilaments, which are ubiquitous in both cancer cells and normal cells. Not surprisingly, disruption of actin microfilaments with cytochalasin D in normal rat embryo fibroblasts cell leads to p53-dependent cell cycle arrest and apoptosis [27]. As the structure of engleromycin is very similar to that of cytochalasin D, we wondered if they share the properties in anti-inflammatory, antivirus, and anticancer effects.

### 2.3. Top II Inhibition Assay In Vitro

Numerous secondary metabolites from *E. goetzei*, such as the analogues of cytochalasins, chaetoglobosins, and aspochalasans, have shown significant inhibitory effects against a variety of tumor cell lines by inhibiting the cytoplasmic cleavage during cell division [28,29,30,31]. However, since the first isolation of engleromycin from *E. goetzei* in 1980 [2], no further studies on the anti-tumor activity of this compound have been reported. Our antiproliferative assays revealed that engleromycin could strongly inhibit the proliferation of tumor cell lines of SGC-7901, HT-29, HeLa and A549. Top II acts on the DNA double strands to cause the transient break, and then opening and reclosing of the double-stranded enzyme bridge, so as to maintain the normal topological state of DNA [15]. Top II is highly expressed in many cancer cells [32,33]. Therefore, Top II has become the very attractive targets for the development of chemotherapeutic drugs [34]. Herein, it’s important to figure out the inhibitory capacity of engleromycin on Top II.

As shown in Table 2, engleromycin exerted strong inhibitory activity on Top II in a dose-dependent manner with the IC_50_ value at 35.43 ± 1.85 µM, but weaker than that of doxorubicin at 4.92 ± 0.28 µM. Other report of etoposide [35], another well-known Top II inhibitor in clinic, also presented lower IC_50_ value at 78.4 µM and further approved our finding in this assay. As a result, it is concluded that engleromycin might be a very promising anticancer drug as the Top II inhibitor.

### 2.4. Molecular Docking Simulation

The Top II inhibitory activity of engleromycin prompted us to carry out the molecular docking simulations to further observe the ligand-protein interactions in detail. As shown in Figure 2a, doxorubicin and engleromycin initially docked into the same active domains of the target enzyme of interest. Additionally, considering that the energy score could reflect the potential bonding ability between drug and its receptor, engleromycin (−16.62 kcal/mol) possessed relatively weaker binding than doxorubicin (−19.85 kcal/mol), according to the docking calculations in Table 2, and which also supported the Top II inhibitory rank of the two aforementioned compounds. 

For the specific docking interaction with Top II, the engleromycin showed 2 hydrogen bond (H-bond, red dashed line) interactions with the peptide linkage (formed between the residues Gly52 and Ser53) and the residue Asp374, and also the van der Waals and hydrophobic interactions with Gln59, Sre320, Lys321 and Glu379 (Figure 2b). For the positive control, doxorubicin shared the same the residues of Gly52, Lys321, Asp374 (Figure 2c) with engleromycin to Top II. Hence, to some extent, the above H-bonds enhanced the potential binding affinities of engleromycin, thereby interfering with or inhibiting the physiological activities of the two target enzymes. Doxorubicin could inhibit the malignant proliferation of tumor cells by interfering with the DNA’s replication, transcription and synthesis, and have exhibited remarkable therapeutic effects on digestive tract cancers in clinic, including colorectal cancer, gastric cancer, esophageal cancer, etc. [36]. From this point, when docking into the same binding pocket in Top II with doxorubicin (Figure 2a), engleromycin might also interfere with the DNA biosynthesis during tumorigenesis. 

## 3. Materials and Methods

### 3.1. General Experimental Procedures

^1^H-, ^13^C-, and 2D-NMR spectra were carried out in CDCl_3_ using a Bruker AM-400 MHz spectrometer (Karlsruhe, Baden-Wuerttemberg, Germany) with TMS as internal standard (IS); δ in ppm and *J* in Hz. ESI-MS was obtained with a TSQ Quantum Access MAX mass spectrometer (Thermo Fisher Scientific, Waltham, MA, USA). Column chromatography was performed using silica gel (200–300 mesh, Qingdao Marine Chemical Ltd., Qingdao, China), RP-C18 silica gel (150–200 mesh, Merck, Darmstadt, Hesse, Germany) and Sephadex LH-20 (20–100 μm, Sigma, St Louis, MO, USA). FBS (fetal bovine serum) and DMEM (Dulbecco’s Modified Eagle Medium) were purchased from Gibco (Life Technologies, Grand Island, NY, USA). Doxorubicin was provided by Sigma-Aldrich (St Louis, MO, USA). All other chemicals and solvents were of analytical grade.

### 3.2. Mushroom Material

The fruiting bodies of fungus *E. goetzei* were collected from Mount Kenya (Kenya), and kindly authenticated by Prof. Guangwan Hu, a taxonomist in Wuhan Botanical Garden, Chinese Academy of Sciences. A voucher specimen was deposited in the herbarium of the Key Laboratory of Plant Germplasm Enhancement and Specialty Agriculture.

### 3.3. Extraction and Isolation

The air-dried fruiting bodies of *E. goetzei* (4.0 kg) were extracted with 80% ethanol by ultra-sonication for 30 min at room temperature with three replications. After removal of the solvent in vacuo, the residue was dispersed in H_2_O and then partitioned successively with petroleum ether (PE) and ethyl acetate (EA). The EA-soluble fraction (20.2 g) was chromatographed on a silica gel (200–300 mesh) column, and eluted with PE/EA (10:0 to 0:1) and EA/MeOH (1:0 to 0:1) to give seven fractions (EG1-EG7). The fraction EG-7 (6.7 g) was further subjected to a MPLC RP-C18 silica gel column with MeOH/H_2_O (0:1 to 1:0) to give a major sub-fraction, which was finally purified by repeated Sephadex LH-20 column with MeOH/H_2_O (98:2) to afford the EG-7h (53 mg).

EG-7h: White amorphous powder. ^1^H-NMR (CDCl_3_) δ_H_: 8.30 (1H, s, H-2-NH), 7.32–7.18 (5H, m, H-2′, 3′, 4′, 5′, 6′), 5.59 (1H, dd, *J* = 15.43, 9.63, H-14), 5.29 (1H, ddd, *J* = 15.47, 10.12, 5.49, H-21-OH), 5.21 (1H, d, *J* = 5.69, H-13), 4.97 (1H, s, H-12A), 4.75 (1H, s, H-12B), 4.38 (1H, d, *J* = 6.36, H-7-OH), 4.32 (1H, s, H-18-OH), 3.49 (1H, dd, *J* = 10.33, 6.25, H-7), 3.38 (1H, d, *J* = 5.66, H-21), 3.29 (1H, m, H-19), 3.15 (1H, m, H-16, 20), 3.07 (1H, s, H-3), 2.72 (1H, dd, *J* = 13.10, 4.68, H-10A), 2.58 (1H, dd, *J* = 13.03, 7.74, H-10B), 2.43 (1H, m, H-5), 2.29 (1H, m, H-4, 15A), 1.94 (1H, m, H-15B), 1.36 (3H, d, *J* = 12.88, H-23), 1.03 (3H, d, *J* = 6.69, H-22), 0.45 (3H, d, *J* = 6.68, H-11). ^13^C-NMR (CDCl_3_) δ_C_: 216.1 (C-17), 175.3 (C-1), 151.3 (C-6), 137.4 (C-1′), 131.4 (C-14), 130.3 (C-13), 130.0 (C-2′, 6′), 128.2 (C-3′, 5′), 126.5 (C-4′), 111.0 (C-12), 76.6 (C-18), 72.9 (C-21), 70.3 (C-7), 59.4 (C-19), 54.9 (C-20), 53.5 (C-9), 52.8 (C-3), 48.5 (C-4), 44.6 (C-8), 43.7 (C-10), 42.0 (C-16), 37.6 (C-15), 31.9 (C-5), 23.0 (C-23), 18.9 (C-22), 12.8 (C-11). ESI-MS *m*/*z* (%): 482.24 [M + H]^+^ (1), 446.25 (28), 428.24 (100), 404.23 (65), 386.22 (65), 344.16 (23), 281.16 (37), 239.17 (31), 120.25 (37), 91.25 (10).

### 3.4. Antiproliferative Assays

Human cancer cell lines were used for antiproliferative assays, and which were maintained in DMEM/high glucose medium supplemented with 10% fetal bovine serum (FBS), 100 U/mL penicillin, and 100 µg/mL streptomycin. These cells were cultured in a cell incubator under humidified conditions with 5% CO_2_ at 37 °C. Cell viability was determined by MTT (3-(4,5- dimethyl-2-thiazolyl)-2,5-diphenyl-2-*H*-tetrazolium bromide) assay as described previously [37]. In brief, cells were seeded into 96-well plates with a density of 1.0 × 10^4^ per well. After 24 h incubation, the culture medium was removed carefully and treated with 100 μL fresh medium containing various concentrations (0.00, 0.77, 2.31, 6.92, 20.76, 62.29 μM) of engleromycin. After 72 h of treatment, 20 μL of MTT solution (5 mg/mL) was added into each well and incubated for 4 h in the cell incubator. The complete medium with less than 0.1% DMSO was used as the normal control; doxorubicin was served as the positive control, with the tested concentrations at 0, 0.03, 0.08, 0.23, 0.68, 2.04, 6.13, 18.4, 55.2 μM. The absorbance was measured by a microplate reader at a wavelength of 490 nm. The cell growth inhibition rate was calculated as follows:% cell inhibition = (mean OD control − mean OD sample)/mean OD control × 100%
where OD control and OD sample are the OD values of control and three samples, respectively. The data were expressed as means ± SD of three replicates.

### 3.5. Top II inhibition Assay In Vitro

DNA Top II inhibitory assay was performed using the protocols reported previously with some modification [37,38]. The reaction mixture (20 μL) contains 10 mM Tris-HCl (pH 7.9), 175 mM KCl, 0.1 mM EDTA, 5 mM MgCl_2_, 2.5% glycerol, 1 mM ATP and DNA. In addition, 2 units of human Top II were applied for each assay. Top II dilution buffer includes 10 mM Sodium phosphate (pH 7.1), 50 mM NaCl, 0.2 mM DTT, 0.1 mM EDTA, 0.5 mg/mL BSA and 10% glycerol. Doxorubicin was used as the positive control, and each sample was carried out in triplicate. The IC_50_ values were determined using the nonlinear regression analysis and their dose-response curves were acquired with the SigmaPlot (V 12.5, Systat Software Inc., San Jose, CA, USA).

### 3.6. Molecular Docking

In order to further discuss the potential mechanism of antiproliferative activity, molecular docking studies were performed by applying the AutoDock Vina (V 1.1.2, National Biomedical Computation Resource, San Diego, CA, USA). With this purpose, the crystal structure of Top II (PDB: 1ZXM) was acquired from the Protein Data Bank (www.rcsb.org). The docking simulation procedures were carried out on base of our previous work with some modification [39,40,41]. Prior to the docking simulation, the hydrogen atoms were added to each structure, and then the structures of Top II and engleromycin were energy-minimized in the MMFF94× force field. Due to the explicitly active residue of Lys378 in the ATPase domain of human Top II [42], the ligand was docked into the putative binding site using the MOE tool to establish applicable conformations. The centroid coordinates within a 10.0 Å radius sphere centered on the active site was served as the docking region, namely, the grid box (coordinates, x: 43.801, y: −0.089, z: 25.764) centered on the active site were setup with a map of 60 × 60 × 60. Later, the docking calculations were carried out with the 2.5 × 10^6^ energy evaluations and 30 independent Genetic Algorithm runs. Finally, all docking results and 3D geometries were observed using the AutoDock Tools and PyMOL Molecular Visualization (V 2.2, Delano Scientific LLC, San Carlos, CA, USA).

## 4. Conclusions

As a long-standing traditional medicine in Africa and China, *E. goetzei* has very good antibacterial, anti-inflammatory and anticancer effects. Engleromycin was firstly isolated from *E. goetzei* in 1980, with its biological activity remaining unknown. To the best of our knowledge, this is the first report on anticancer activity of engleromycin. Antiproliferative experiments of human cancer cells like HT-29, SGC-7901, HeLa and A549 demonstrated that engleromycin possess potent anticancer activity, which was further confirmed by molecular docking analysis using target enzymes Top II and Top II inhibition assay. Future investigations are necessary to explore the underlying mechanism of anticancer activity of engleromycin, including but not limited to the effect of engleromycin on cancer cell cycle, apoptosis, and autophagy.

## Figures and Tables

**Figure 1 molecules-24-00166-f001:**
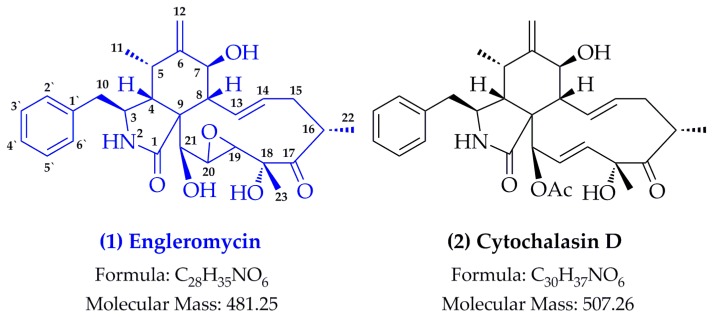
Structures of engleromycin and cytochalasin D.

**Figure 2 molecules-24-00166-f002:**
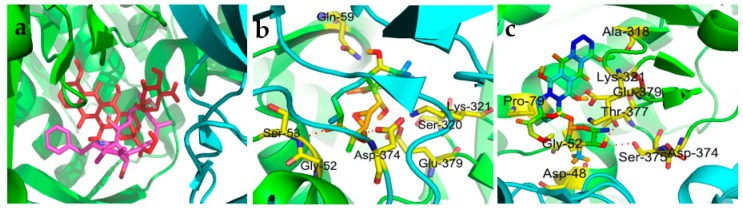
Molecular docking simulations of the overlay view of engleromycin (magenta) with doxorubicin (red) in the Top II (PDB: 1ZXM) binding pocket (**a**), and their ligand–protein interactions between engleromycin (**b**) and doxorubicin (**c**) with Top II, respectively.

**Table 1 molecules-24-00166-t001:** Antiproliferative effects and cytotoxicity of engleromycin and doxorubicin.

Compound	IC_50_ (µM) ^a^				
	SGC-7901	HT-29	HeLa	A549	LO-2
Engleromycin	26.77 ± 1.69	7.73 ± 0.18	7.00 ± 0.12	3.14 ± 0.03	3.76 ± 0.29
Doxorubicin	0.74 ± 0.03	0.24 ± 0.02	0.27 ± 0.02	0.14 ± 0.01	0.58 ± 0.03

^a^ MTT method, SGC-7901, HT-29, HeLa, A549 and LO-2 cells incubated with the compounds for 72 h. Doxorubicin was used as the positive control.

**Table 2 molecules-24-00166-t002:** Top II inhibitory activities and energy scores of engleromycin and doxorubicin.

Compound	Inhibition of Top II (µM)	Energy Score (kcal/mol)
Engleromycin	35.43 ± 1.85	−16.62
Doxorubicin	4.92 ± 0.28	−19.85
Etoposide [3]	78.4

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
