# Peer review of "Antiproliferative and Enzyme Docking Analysis of Engleromycin from Engleromyces goetzei"

_molecules, 2019, doi:10.3390/molecules24010166_

Round 1

Reviewer 1 Report

Zhang Y. et al. have done the major requested revisions, particularly concerning the biological tests with supplementary cancer cell lines for the cytotoxicity studies and evaluation of Top II inhibition.

-The manuscript must be thoroughly still checked for English style and misspellings (examples: line 45, reports; line 57, prevents the Top II binds to DNA; line 59, targets on top II through stabilize; line 93: its analogue of….)

-Minor modifications: line 61: its structure was confirmed (not elucidated since it was already published, ref 2); lines 66-67: suppress the unuseful last sentence; line 96: 2.2 subtitle: complete the cell lines, cell growth of SGC-7901, HT-29, HeLa, A549 and LO-2

-Major modifications: it is a nonsense to present 5-FU, an antimetabolite compound as a potential Top II inhibitor unless the authors bring some bibliographic proofs… In this context, in the abstract 5-FU must be deleted; it should also be removed from Table 2 (the topII IC50 value of 116 mM is not surprising). The discussion part (lines 133-139) should be reformulated; engleromycin with an IC50 value of 35 mM is not comparable to doxorubicin (4.9 mM); is is 7-fold less inhibitory. In the 2.4. docking part; 5-FU should be removed and the comparison should be limited to engleromycin and doxorubicin. In Figure 2’s legend, please add the TopII PDB code.

In Table 1, the cytotoxic data of doxorubicin on HT-29, HeLa, A549 and Lo-2 should be added since it is the reference compound as TopII inhibitor. These data would perfectly complete paragraph 2.2. and the appended discussion.

Author Response

Comments and Suggestions for Authors

Zhang Y. et al. have done the major requested revisions, particularly concerning the biological tests with supplementary cancer cell lines for the cytotoxicity studies and evaluation of Top II inhibition.

-The manuscript must be thoroughly still checked for English style and misspellings (examples: line 45, reports; line 57, prevents the Top II binds to DNA; line 59, targets on top II through stabilize; line 93: its analogue of….)

Response: We thank the reviewer`s advice in this regard and have thoroughly revised the relevant English styles and misspellings in the manuscript.

-Minor modifications: line 61: its structure was confirmed (not elucidated since it was already published, ref 2); lines 66-67: suppress the unuseful last sentence; line 96: 2.2 subtitle: complete the cell lines, cell growth of SGC-7901, HT-29, HeLa, A549 and LO-2

Response: We thank the reviewer`s advice and have revised the description as suggested in the manuscript.

-Major modifications: it is a nonsense to present 5-FU, an antimetabolite compound as a potential Top II inhibitor unless the authors bring some bibliographic proofs… In this context, in the abstract 5-FU must be deleted; it should also be removed from Table 2 (the top II IC50 value of 116 mM is not surprising). The discussion part (lines 133-139) should be reformulated; engleromycin with an IC50 value of 35 mM is not comparable to doxorubicin (4.9 mM); is is 7-fold less inhibitory. In the 2.4. docking part; 5-FU should be removed and the comparison should be limited to engleromycin and doxorubicin. In Figure 2’s legend, please add the TopII PDB code.

Response: We thank the reviewer`s advice and have revised the all descriptions as suggested in the context.

In Table 1, the cytotoxic data of doxorubicin on HT-29, HeLa, A549 and Lo-2 should be added since it is the reference compound as TopII inhibitor. These data would perfectly complete paragraph 2.2. and the appended discussion.

Response: We thank the reviewer`s advice and have added the cytotoxic data of doxorubicin on HT-29, HeLa, A549 and LO-2 as suggested in Table 1.

Reviewer 2 Report

This work describes the characterisation of a fungal-derived potential anti-cancer agent. While this reviewer cannot speak effectively to the experimental aspects of the work, there are two issues that the authors should consider addressing in the computational part of the work: - The authors demonstrate that energies from AutoDock loosely correlate with experimental inhibition studies for the three examined molecules. The authors should consider expanding this analysis (perhaps using AutoDock Vina, rather than AutoDock, for greater speed of calculation and improved accuracy) to all molecules tested against the target (https://www.ebi.ac.uk/chembl/target/inspect/CHEMBL2094255) to comprehensively verify that the docking scores correlate with experimental IC50s in this case. It may be appropriate to divide the molecules into training and test sets for this purpose. - Secondly, and more importantly, the structure preparation and docking procedure employed for examining a molecule with a large ring structure such as engleromycin is inappropriate. As AutoDock does not perform ring sampling, rings must be sampled prior to docking calculations. The authors should perform an extensive conformational search on engleromycin to identify low energy ring conformations and dock one or more of these structures instead. This is likely to change the docking score reported in Table 2 and may result in the loose correlation with experiment no longer applying, hence the importance of carrying out the first suggested work.

Author Response

This work describes the characterisation of a fungal-derived potential anti-cancer agent. While this reviewer cannot speak effectively to the experimental aspects of the work, there are two issues that the authors should consider addressing in the computational part of the work: - The authors demonstrate that energies from AutoDock loosely correlate with experimental inhibition studies for the three examined molecules. The authors should consider expanding this analysis (perhaps using AutoDock Vina, rather than AutoDock, for greater speed of calculation and improved accuracy) to all molecules tested against the target (https://www.ebi.ac.uk/chembl/target/inspect/CHEMBL2094255) to comprehensively verify that the docking scores correlate with experimental IC50s in this case. It may be appropriate to divide the molecules into training and test sets for this purpose. - Secondly, and more importantly, the structure preparation and docking procedure employed for examining a molecule with a large ring structure such as engleromycin is inappropriate. As AutoDock does not perform ring sampling, rings must be sampled prior to docking calculations. The authors should perform an extensive conformational search on engleromycin to identify low energy ring conformations and dock one or more of these structures instead. This is likely to change the docking score reported in Table 2 and may result in the loose correlation with experiment no longer applying, hence the importance of carrying out the first suggested work.

Response: We thank the reviewer`s advice and used AutoDock Vina for further energy scores. Prior to the docking, the structures of engleromycin and Top II were both energy-minimized in the force field for the lowest energy ring conformations. For the docking study, we applied the flexible docking produces for the ring structure of engleromycin. At this point, engleromycin was docked against the ATPase domains of Top II in MOE that uses a simulated annealing search protocol to find favorable binding conformations between a small, flexible ligand and a rigid target protein. Following the above procedures as suggested, the modified docking scores were calculated again and shown in Table 2.

Reviewer 3 Report

The paper by Zhang and coworkers deals with the analysis of antiproliferative activity and docking study of Engleromycin on Topoisomerase II. Engleromycin has interesting properties and its activity is not yet well understood, so it deserves studies to be better elucidated. On the whole, the paper is good and sound, but  the docking part would need some improvement before publication.

In particular, the choice of the site were the molecules (Engleromycin and controls) have been docked is not well justified in the text. Autodock requires the definition of a docking grid that is centered on the putative binding site. How was the position of this grid (and so the site) chosen? Are there any information in literature, or a preliminary blind docking has been performed? Such details would be very important for the interested reader. 

Referring to the literature on molecular docking on topisomerase II (e.g. Jadhay et al. In Silico Pharmacol. 2017; 5: 4, Christodoulou et al. European journal of medicinal chemistry, 92, 766, Rani et al. American Journal of Pharmacological Sciences, 2014, 2, 42 ecc. ) could help in getting the general picture in which the paper is inserted. 

Moreover, figure 2 is not very readable, the color code of the molecules and of the protein do not allow easy identification of crucial feathures. 

In general, I recommend the publication of the paper after minor revisions. 

Author Response

The paper by Zhang and coworkers deals with the analysis of antiproliferative activity and docking study of Engleromycin on Topoisomerase II. Engleromycin has interesting properties and its activity is not yet well understood, so it deserves studies to be better elucidated. On the whole, the paper is good and sound, but the docking part would need some improvement before publication.

In particular, the choice of the site were the molecules (Engleromycin and controls) have been docked is not well justified in the text. Autodock requires the definition of a docking grid that is centered on the putative binding site. How was the position of this grid (and so the site) chosen? Are there any information in literature, or a preliminary blind docking has been performed? Such details would be very important for the interested reader.

Response: We thank the reviewer`s advice in this regard. For better elaboration for the interested readers, in the revised manuscript, we have added the information of the putative binding site in the ATPase domain of human Top II and coordinates of the grid box in the section “3.6 Molecular Docking”, according to several previous literatures as suggested.

Referring to the literature on molecular docking on topisomerase II (e.g. Jadhay et al. In Silico Pharmacol. 2017; 5: 4, Christodoulou et al. European journal of medicinal chemistry, 92, 766, Rani et al. American Journal of Pharmacological Sciences, 2014, 2, 42 ecc. ) could help in getting the general picture in which the paper is inserted.

Response: We thank the reviewer`s advice in this regard. We not only have rewritten some docking procedures in the section “3.6 Molecular Docking”, and also renewed the references of molecular docking on topisomerase II in the context as suggested.

Moreover, figure 2 is not very readable, the color code of the molecules and of the protein do not allow easy identification of crucial features.

In general, I recommend the publication of the paper after minor revisions.

Response: We thank the reviewer`s advice in this regard.

Round 2

Reviewer 1 Report

The authors have brought all the expected revisions

This manuscript is a resubmission of an earlier submission. The following is a list of the peer review reports and author responses from that submission.

Round 1

Reviewer 1 Report

Zhang Y. et al. describes in their paper the "Antiproliferative and Enzyme Docking Analysis of Engleromycin".

This paper lacks deeply of originality since Engleromycin is structurally very close to the previously reported compound, cytochalasin D and its structure elucidation was already done (Tet Lett 1980, 21, 5079). So the "Structure  Elucidation" paragraph is not useful.

The biological data in this paper are very poor and only based on the inhibition of cell growth of two cancer cell lines. More extensive studies on other cancer cell lines and on a normal cell line should be done to consolidate the status of engleromycin as an effective anticancer agent. Is it pro-apoptotic?... Another positive reference like Paclitaxel shlould be used instead of 5-FU...

Given the data, the docking study on Topo II and Thymidylate Synthase is much too premature since no firm experimental proof for such enzymatic inhibitions was brought... This should be done first...

Reviewer 2 Report

Authors of the manuscript "Antiproliferative and Enzyme Docking Analysis of 2 Engleromycin from Engleromyces goetzei" have presented work on the determination of anticancer activity of a previously isolated natural product, engleromycin. The authors have attempted to propose a putative mechanism of action  by using a molecular docking against two validated cancer targets.

This work may be of potential interests to readers due to their attemt to  report the  anticancer activity of engleromycin for the first time, however, it is not surprising that this molecule is active as the close structural analogue cytochalasin D exerts a number of different biological activities as authors have already recognized.

However, this manuscript has several weaknesses that needs addressing:

- the spectral data are not shown, which should be enclosed at least in the supplemetary material;

- authors state that "EG-7h was 82 engleromycin, which exhibited very similar 1H and 13C data and isolated from the same fungus", however they do not provide an explanation why there are differences between previously reported data, especially when considering that peaks have significantly different chemical shifts;

- authors state IC50 of 5-FU and adriamycin, but do not provide relevant information in the methods. Is that value previosly reported or authors conducted  the experiment but failed to add information in the methods section;

- the choice of the protein targets for the docking analysis is superficial. Authors have selected only two validated targets, however there are far more proteins that could be possible targets   for this molecule. Authors must include docking results against more comprehensive panel of proteins, or pprovide an experimental evidence that engleromycin binds or inhibits two  selected proteins;

- the analysis and discussion of the docking results should be improved, as there are no quantitative information on interaction energies and comparison of those  with  positive controls.

- authors did not provide an explanation of the fact that the binding residues for engleromycin and positive controls are different, despite visually being in same binding sites?

- the details of the docking methods are provided. Authors refer to a previously published procedure and have mentioned that there are modifications. however, no modifications were detailed. This is important especiallu as the protein targets are different and therefore the binding site centers are  different. Authors also need to provide information on how they have dealt with missing residues in the protein targets.

- authors state that engleromycin is potentially a candidate for further preclinical trials, however, there are no information on toxicity against healthy human cells and authors did not provide any insights on drug-like properties of engleromycin.

Furthermore, the structure of the manuscirpt is not adequate, with docking results writen in a completely different style to the rest of the paper.  Some of the information from the section 2.3 probably should be in the introduction or discussion.

The manuscript needs proofreading as there are omissions that needs to be addressed. Here are some examples of the problem, but this is not a comprehensive list.

Line 39
In China, E. sinensis has  a wide range of biological activities, such as anti-inflammatory, anti-microbial, anti-cancer, anti-virus properties as well as cholesterol ester transfer protein (CETP) inhibition activity

Line 65

For the first time, the antiproliferative activity of engleromycin and its potential mechanisms were conducted in this paper.

Line 80

Additionally, the complete assignments of proton and carbon signals in the structure were further confirmed by using 2D-NMR techniques, such as 1H-1H COSY, HMBC and HMQC spectra.

Line 97

Based on the results of Figure 1, the spectrum of engleromycin is close to being identical to cytochalasin D

Line 169

1H-, 13C-, and 2D-NMR spectra were carried out in CDCl3 using a Bruker AM-400 MHz spectrometer with TMS as internal standard (IS); δ in ppm and J in Hz

and many others.

Therefore, manuscript in this form is not suitable for publishing in Molecules.

Reviewer 3 Report

The paper is of average quality, the following issues have to be improved in order to be published:

English should be improved throughout the manuscript i.e. line 97 “is close to being identical”, etc.

In introduction please describe topoisomerase inhibitors and thymidylate kinase inhibitors and the resemblance in terms of structural features with engleromycin, or mention if one of the compounds you make reference to in this paper are known as inhibitors of topoisomerase or thymidylate kinase. 

Pag. 3 line 97 the sentence : “Based on the results of Figure 1, the spectrum of engleromycin is close to being identical to  cytochalasin D [2] should be reformulated since structural similarity, not spectrum is driving analogous biological activity. Moreover, if previous publication by Pedersen et al suggested that you should mention explicitly this.

Pag. 3 line 103 “Given that engleromycin shares similar structure with cytochalasin D, we have reasons to speculate that engleromycin could have anti-inflammatory, antivirus, and anticancer effects.”

Do cytochalasin D and 19,20-epoxycytochalasin D are inhibitors of Topoisomerase II a and thymidylate kinase? If yes, please mention this in the paper.

Docking methodology description should be done in more detail.

The docked poses in Figure 2 are selected based on energetic criteria?

Docking into thymidylate kinase structure – without ligand – is not convenient.

Comment the position of engleromycin, why do you think is different?

What about the interactions of engleromycin with receptor? Are they being similar to the interactions observed in co-crystal of Topo II with AMP-PNP or ADP?

Conclusions: Docking results itself cannot confirm that engleromycin is an inhibitor of Topo II and TS. If possible in vitro experiments can confirm this, if not then delete this sentence from Conclusions.